# Where does an LLM begin computing an instruction?

**Aditya Pola**
Indian Institute of Technology, Hyderabad

**Vineeth N. Balasubramanian**
Indian Institute of Technology, Hyderabad

## Abstract

Following an instruction involves distinct sub-processes, such as reading content, reading the instruction, executing it, and producing an answer. We ask where, along the layer stack, instruction following begins—the point where reading gives way to doing. We introduce three simple datasets (Key–Value, Quote Attribution, Letter Selection) and two hop compositions of these tasks. Using activation patching on minimal-contrast prompt pairs, we measure a layer-wise flip rate that indicates when substituting selected residual activations changes the predicted answer. Across models in the Llama family, we observe an inflection point, which we term onset, where interventions that change predictions before this point become largely ineffective afterward. Multi-hop compositions show a similar onset location. These results provide a simple, replicable way to locate where instruction following begins and to compare this location across tasks and model sizes.

## 1 Introduction

Instruction following is central to how language models are used. While most studies [Ouyang et al., 2022, Chung et al., 2024, Wang et al., 2023, Zheng et al., 2023, Zhou et al., 2023] ask whether a model follows an instruction, orthogonally, we aim to find the location of this instruction-following behaviour. In transformers, computation proceeds layer by layer with each block writing additively to a shared residual stream that later blocks read, so it is natural to ask at what depth instruction information first begins to affect predictions [Elhage et al., 2021].

Depth matters for instruction processing. [Olsson et al., 2022] identifies induction circuits that emerge at characteristic layers and implement copy-style adaptation from the prompt. Causal interventions for factual recall [Meng et al., 2022] localize mediators in mid-network components and show that behavior can be edited without retraining. Studies trace how factual and counterfactual behaviors dominate at different depths [Ortu et al., 2024]. Analyses of chain of thought [Chen et al., 2025] report depth-dependent structure in attention and feature pathways that support stepwise reasoning. [Templeton et al., 2024, Lan et al., 2025] extract interpretable features with sparse autoencoders and find organization aligned with layer depth. Open problems in mechanistic interpretability emphasize identifying the roles of network components and using that understanding to control and predict behavior [Sharkey et al., 2025].

Our contributions are two-fold. First, we introduce three small, purpose-built datasets (Key-Value, Quote Attribution, and Letter Selection) together with two-hop versions that increase procedural complexity while keeping answers unambiguous. Second, we identify where instruction following begins inside the network, understood as the depth at which the model shifts from reading the prompt to carrying out the intended operation. Across Llama models this start-of-execution location is consistent and remains similar in the two-hop setting, providing a clear reference point for comparing models and tasks and guiding diagnostics and model design.

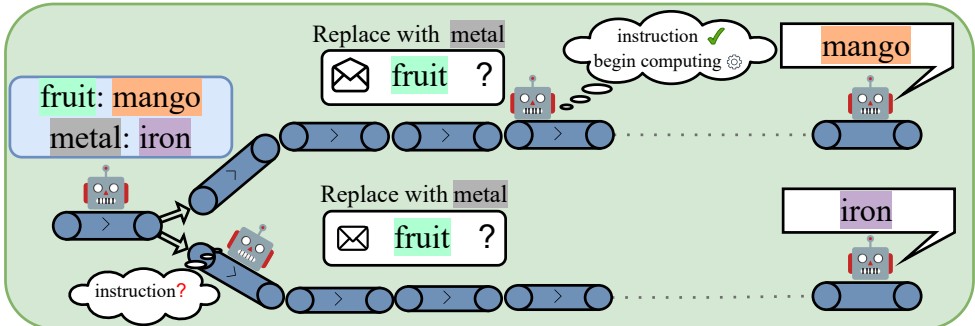

Figure 1: **Overview: timing of instruction replacement determines the outcome.** In a key–value prompt (CONTENT → `fruit:mango`, `metal:iron`; INSTRUCTION KEYWORD → `fruit?`), the instruction box shows *unread/read* icons to indicate whether the model has already read the instruction at the time of the intervention. The intervention is shown as the text "replace with `metal`." In the upper path, the replacement appears *after* the instruction is read, and the answer remains `mango`. In the lower path, the replacement appears *before* it is read, and the answer flips to `iron`. Timing the replacement reveals when instruction following begins.

## 2  Setup

### 2.1  Datasets

We begin by creating three new dataset for our analysis. Our datasets were made considering the following: *(a)* the prompt should contain all the information required to answer a question, without requiring any external knowledge (this is to avoid any confounding from the parametric memory of the model), *(b)* each task must allow for one counterfactual that results in an answer flip (this enables us to perform causal mediation analysis), *(c)* the instruction needs to be token aligned between the original and the counterfactual prompt (allowing us to automate the patching experiments).

Each prompt has three segments: *content* (facts needed to answer), *instruction* (what to do with the content), and a *format directive* (e.g., "answer in one word"). Only the *instruction tokens* differs across each contrastive pair; the content and format directive are held fixed. The counterfactual instruction is a minimally changed instruction that reverses the correct label under a clean run, ensuring a potential answer flip.

We introduce three datasets in line with the above requirements: *Key-value* requires the retrieval of the *instruction* specified key and return it's value. In *Quote-Attribution*, the model is tasked with returning the name of the person that wants to go to the specified location in the *instruction*. *Letter-selection* tests a models ability to return the first or last letter of a given keyword. Further details and examples can be found in the appendix [A].

#### 2.1.1  Multi-hop variants

To mitigate confounding from task complexity, we construct *two-hop* compositions. Each instance is an ordered pair of *distinct* base tasks drawn from {Key–Value, Quote Attribution, Letter Selection}. The second hop's query is a deterministic function of the first hop's output, enforcing strict serial dependence; thus the final answer is obtainable only if hop-1 is correctly solved.

## 3  Methodology

We perform our experiments on the Llama [Grattafiori et al., 2024] family and analyze scaling effects across model size, using Llama3.2-1B-Instruct, Llama3.2-3B-Instruct, and Llama3.1-8B-Instruct.

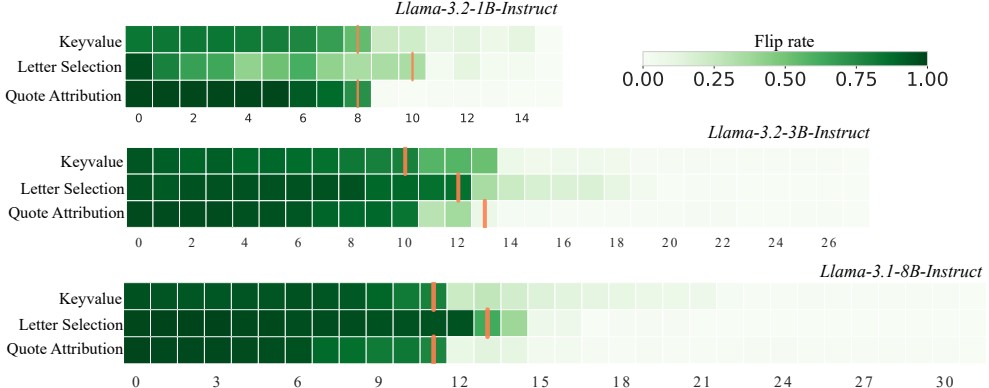

Figure 2: **Single-hop keyword-span causal patching.** Panels plot layer-wise *flip rate*—the fraction of items whose answer switches when substituting residuals at the keyword tokens—against transformer layer; the vertical orange bar marks the *changepoint* (instruction onset). Across Key–Value, Letter Selection, and Quote Attribution, and across Llama3.2-1B/3B and Llama3.1-8B, flip rate is high from early layers up to onset and then approaches baseline. This indicates a depth-localized boundary: after onset, replacing keyword residuals rarely affects the answer, whereas before onset it often does. Onset depth varies across tasks and models, but the pre/post contrast is consistent.

## 3.1 Activation Patching

We use activation (causal) patching [Heimersheim and Nanda, 2024] on the *residual stream*. Given a base prompt $B$ and a source prompt $S$ that differ only in the instruction span, we first cache clean activations. For layer $\ell$ and token mask $\mathcal{M}$, we modify the state by substituting residual activations from $S$ into $B$ at the masked positions and resuming the forward pass on $B$:

$$\tilde{\mathbf{h}}_i^{(\ell)} = \begin{cases} \mathbf{h}_i^{(\ell)}(S), & i \in \mathcal{M}, \\ \mathbf{h}_i^{(\ell)}(B), & i \notin \mathcal{M}. \end{cases}$$

All subsequent computations use $\tilde{\mathbf{h}}^{(\ell)}$ as the residual stream for $B$.

**Spans.** *Keyword* — the minimal trigger words inside the instruction. Patching tests whether these tokens are *behaviorally decisive* i.e., substituting their residuals is sufficient to flip the model's answer.

*Content-control* — position/length-matched tokens from the content segment that are *identical* in $B$ and $S$. This is a negative control to isolate *position/routing artifacts*: flips driven by token position, generic attention-routing patterns, or activation-scale effects rather than instruction-specific content.

*Answer bottleneck* — the final context token (the last token before the model emits its first output token). Patching here probes *late consolidation*: whether the model compresses and places the instruction signal at this token immediately prior to decoding.

**Metric.** We report the *flip rate* (a.k.a. *intervention success rate*) the fraction of items whose predicted label switches from the base to the source under the patched span:

$$\text{FlipRate}_{\ell,\mathcal{M}} = \frac{1}{N} \sum_{n=1}^{N} \mathbf{1}\left[\arg\max_{y} p\big(y \mid \text{patched}_{\ell,\mathcal{M}}^{(n)}\big) = y_S^{(n)}\right].$$

**Onset via change-point analysis.** For each layer-wise flip-rate curve, we fit a single-break, two-mean step model by minimizing the within-segment sum of squared errors over all split points, imposing at least two layers per segment [Truong et al., 2020]. We report the boundary index (the layer on the left side of the break) as the instruction-onset layer and the mean flip rates on each side.

## 4 Results

In single-hop settings, keyword-span patching exhibits a consistent profile. Figure 2 shows that flip rate is high from the earliest layers up to the onset layer and then falls toward near-zero, indicating

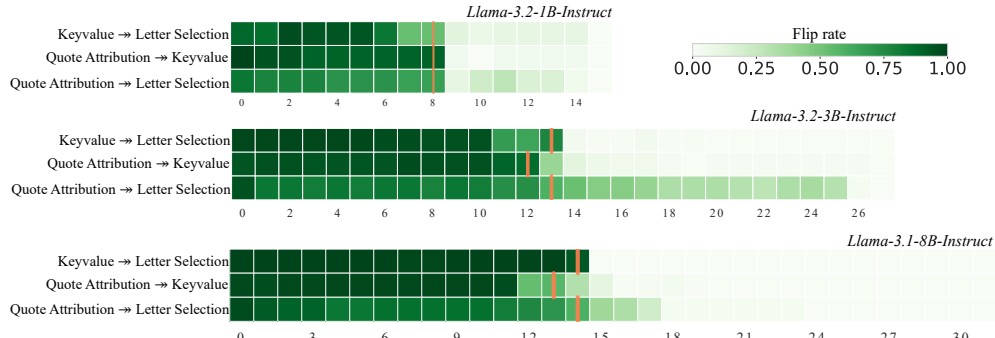

Figure 3: **Multihop keyword-span causal patching.** Layer-wise *flip rate* under residual-stream patching of the *keyword* span for two-hop compositions; the vertical orange bar marks the *changepoint* (instruction onset). Across models and hop orderings, curves remain elevated up to onset and then taper toward baseline, paralleling single-hop behavior. The onset depth matches single-hop (no systematic deeper shift), indicating that added procedural complexity leaves instruction localization unchanged and that post-onset computation proceeds without further reliance on the keyword tokens.

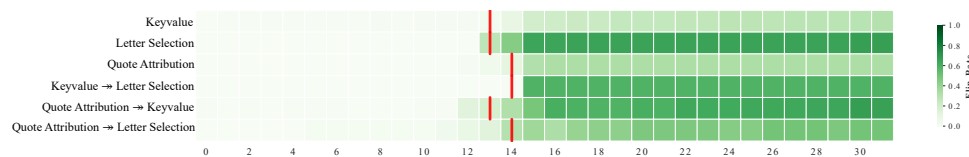

Figure 4: **Answer-bottleneck flip-rate heatmap (Llama3.1–8B-Instruct).** Heatmap of layer-wise *flip rate* when patching the final context token; red vertical lines mark the *changepoint*. Flip rates rise after onset and stay below keyword-span levels, indicating late consolidation of the instruction signal onto the last context token with weaker causal leverage than the original keyword tokens.

a sharp handoff: before onset, swapping keyword residuals is behaviorally decisive, whereas after onset it no longer changes the answer. This pattern is consistent with the model having parsed and internalized the instruction by onset, so subsequent layers do not *re-read* the keyword tokens, as reflected in the near-zero post-onset flip rates.

For multi-hop compositions under *keyword* patching (see Fig. 3), the layer-wise flip-rate curves mirror the single-hop pattern. Across models, the onset occurs at approximately the same layer as in single-hop, with no systematic shift earlier or later despite the added hop. Increased procedural complexity does not move instruction localization deeper; by the onset layer the model has already formed the instruction-specified task policy, and subsequent layers execute that policy without revisiting the keyword tokens.

Figure 4 shows that, for Llama-3.1-8B, patching the answer bottleneck (last context token) yields near-zero flip rates before the red onset line, a delayed rise after onset. This pattern indicates that replacing only the final-token residual becomes behaviorally consequential shortly after onset (consistent with a late handoff to that position), while the peak flip rates remain below those from keyword-span patching, suggesting a weaker, more compressed influence at the bottleneck than at the raw instruction tokens. Note, *content-control* patching results in a zero flip-rate across layers.

## 5 Conclusion

We identify the layer where instruction execution begins. Across Llama models and tasks, flip rate curves show a clear onset; beyond this layer, keyword patches rarely change the answer, and two-hop variants place the onset at a similar depth. Effects at the answer bottleneck appear only after onset and are weaker. Together, these results pin down where instruction execution is established and turn it into a measurable coordinate for benchmarking and targeted analysis.

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

# A Dataset Structures and Examples

**Format directive.**  Examples use the format directives as present in each prompt.

## A.1 Single-hop: Key–Value

**Task.** Return the value associated with a specified key.

**Structure**



**Content:** $key_a$: $value_a$,    $key_b$: $value_b$
**Instruction:** Return the value of $key_a$
**Counterfactual Instruction:** Return the value of $key_b$



**Original prompt.**

```
glyph:  sigma
fruit:  orange
rule:  return exactly the value for the requested field
field:  [INSTR]glyph[/INSTR]
Final answer only:  one word.
Answer:  sigma
```

**Counterfactual prompt.**

```
glyph:  sigma
fruit:  orange
rule:  return exactly the value for the requested field
field:  [INSTR]fruit[/INSTR]
Final answer only:  one word.
Answer:  orange
```

## A.2 Single-hop: Quote Attribution

**Task.** Return the name of the person who wanted a given place.

**Structure**



**Content:** $person_a$ said "let's go to $place_a$," whereas $person_b$ said "let's go to $place_b$"
**Instruction:** Who wanted to go to $place_a$?
**Counterfactual Instruction:** Who wanted to go to $place_b$?



**Original prompt.**

```
context:  jack said "let's go to the park," but ben said "let's go to the
river."
question:  who wanted to go to the [INSTR]park[/INSTR]?
Final answer only:  one word, lowercase.
Answer:  jack
```

**Counterfactual prompt.**

```
context:  jack said "let's go to the park," but ben said "let's go to the
river."
question:  who wanted to go to the [INSTR]river[/INSTR]?
Final answer only:  one word, lowercase.
Answer:  ben
```

## A.3 Single-hop: Letter Selection

**Task.** Return the first or last letter of a keyword.

**Structure**

**Content:** Consider the word: $keyword$
**Instruction:** What is the $first$ letter of the given word?
**Counterfactual Instruction:** What is the $last$ letter of the given word?

**Original prompt.**

```
context:  consider the word "moonlight".
rule:  return the letter indicated by selector
selector:  [INSTR]first[/INSTR]
Final answer only:  one word, lowercase.
Answer:  m
```

**Counterfactual prompt.**

```
context:  consider the word "moonlight".
rule:  return the letter indicated by selector
selector:  [INSTR]last[/INSTR]
Final answer only:  one word, lowercase.
Answer:  t
```

## A.4   Two-hop: KV → Letter

**Task.** Select a field from a record, take its value, then return the first (or last) letter of that value. Only the selector token differs across the contrastive pair.

**Structure**

**Content:** record with fields $\{key_1\colon value_1,\ key_2\colon value_2\}$; rule: take the value of the selected field, then return its first/last letter
**Instruction:** select: $key_1$
**Counterfactual Instruction:** select: $key_2$

**Original prompt.**

```
record:
  color:  indigo
  animal:  camel
rule:  take the value of the selected field, then return its first letter
select:  [INSTR]color[/INSTR]
Final answer only:  one word, lowercase.
Answer:  i
```

**Counterfactual prompt.**

```
record:
  color:  indigo
  animal:  camel
rule:  take the value of the selected field, then return its first letter
select:  [INSTR]animal[/INSTR]
Final answer only:  one word, lowercase.
Answer:  c
```

## A.5   Two-hop: Quote → KV

**Task.** Choose the person who wanted the target place from a quoted context, then return only that person's attribute from a roster (for example, a pet). Only the target place token differs.

**Structure**

**Content:** quoted context with two people and two places; roster mapping person → attribute
**Instruction:** target: $place_a$
**Counterfactual Instruction:** target: $place_b$

**Original prompt.**

```
context:  ava said "let's go to the mall," but omar said "let's go to the
beach."
roster:
  ava:  pet=cat
  omar:  pet=dog
task:  choose the person who wanted the target place; return only that
person's pet.  do not include names.
target:  [INSTR]mall[/INSTR]
Final answer only:  one word, lowercase.
Answer:  cat
```

**Counterfactual prompt.**

```
context:  ava said "let's go to the mall," but omar said "let's go to the
beach."
roster:
  ava:  pet=cat
  omar:  pet=dog
task:  choose the person who wanted the target place; return only that
person's pet.  do not include names.
target:  [INSTR]beach[/INSTR]
Final answer only:  one word, lowercase.
Answer:  dog
```

## A.6 Two-hop: Quote → Letter

**Task.** Choose the person who wanted the target place, then return only the first (or last) letter of that person's name. Only the target place token differs.

**Structure**

**Content:** quoted context with two people and two places; rule to return first/last letter of the selected person's name

**Instruction:** target: $place_a$

**Counterfactual Instruction:** target: $place_b$

**Original prompt.**

```
context:  leo said "let's go to the river," but paula said "let's go to the
library."
task:  choose the person who wanted the target place; return only the first
letter of that person's name.
target:  [INSTR]river[/INSTR]
Final answer only:  one word, lowercase.
Answer:  l
```

**Counterfactual prompt.**

```
context:  leo said "let's go to the river," but paula said "let's go to the
library."
task:  choose the person who wanted the target place; return only the first
letter of that person's name.
target:  [INSTR]library[/INSTR]
Final answer only:  one word, lowercase.
Answer:  p
```

# B  Limitations and Future Work

Our tests use small, controlled datasets. They isolate instruction execution but do not capture the breadth of real tasks, longer contexts, or heterogeneous formatting. Future work should assess whether the same depth patterns hold on natural datasets and varied prompt styles.

We do not study questions that depend on the model's world knowledge. Tasks that hinge on factual recall, long-term memory, or retrieval may shift where instruction execution becomes effective. Evaluating knowledge-heavy prompts and retrieval-augmented setups is an important direction for future work.

We avoid chain-of-thought and use single-token answers. This leaves open how complex instructions are decomposed, whether sub-instructions are processed in parallel or sequentially, and how multi-step reasoning changes the depth profile. Extending the method to multi-token reasoning traces and richer supervision can test these possibilities.

