# OpenReview forum: "Where does an LLM begin computing an instruction?"
_NeurIPS.cc/2025/Workshop/UniReps — UniReps2025_

### Official Review · Reviewer_SVLZ · 2025-09-09
**Promises and Limitations in Studying Cognitive Transitions in Transformers**

**Confidence:** 4

**Review:**

# Summary

This paper investigates the precise computational transition point in Large Language Models where instruction comprehension shifts to execution. The authors introduce a systematic methodology using activation patching on minimal contrast prompt pairs to identify the ``onset'' layer---the point where interventions that previously changed model responses become ineffective. Testing across three simple tasks (Key-Value retrieval, Citation Attribution, Letter Selection) and their two-step compositions on Llama models (3.2-1B, 3.2-3B, 3.1-8B Instruct), they demonstrate a consistent breakpoint pattern. The key finding reveals that the onset occurs at approximately the same layer on single-step and multi-steps tasks, suggesting that models form complete task policies before execution rather than processing instructions step-by-step. The work provides a framework for localizing cognitive transitions in neural language models with clear replication methodology.

# Strengths and Weaknesses

## Strengths

### Methodological Innovation and Rigor

They introduce a novel and systematic approach to a fundamental question in LLM interpretability. The activation patching methodology is appropriately chosen and consistently applied. The experimental design with minimal contrast pairs (e.g., ``Where does Mar\'{\i}a live?'' vs ``Where does Pedro live?'' in identical contexts) effectively isolates the cognitive phenomenon of interest. The flip rate metric
\begin{equation}
\text{FlipRate}_{\ell,\mathcal{M}} = \frac{1}{N} \sum_{n=1}^{N} \mathbf{1}\left[\arg\max_y p\left(y \mid \text{patched}_{\ell,\mathcal{M}}^{(n)}\right) = y_S^{(n)}\right]
\end{equation}
provides a clear, interpretable measure of intervention effectiveness.

### Robust Cross-Task and Multi-Hop Validation

The consistency of findings across different task types strengthens the generalizability within the tested domain. Most significantly, the multi-hop results provide crucial evidence that ``increased procedural complexity does not move instruction localization deeper; by the onset layer the model has already formed the instruction-specified task policy.'' This finding challenges intuitive expectations and suggests the onset captures something fundamental about instruction processing rather than task-specific artifacts.

### Reproducible Framework

They provide sufficient methodological detail for replication, including clear operational definitions of onset identification through change-point analysis using single-break, two-mean step models. The framework enables systematic comparison across models and tasks, filling an important gap in interpretability methodology.

### Clear Experimental Controls

The inclusion of content-control conditions (position/length-matched tokens from identical content segments) effectively isolates position/routing artifacts from instruction-specific effects, strengthening causal claims about the observed phenomena.

## Weaknesses

### Severely Limited Model and Architecture Coverage

Despite framing as a study of ``LLMs,'' the evaluation is restricted to only three instruction-tuned models within a narrow parameter range (Llama3.2-1B-Instruct, Llama3.2-3B-Instruct, Llama3.1-8B-Instruct). This represents an extremely limited sample that cannot support broad claims about language models generally. More critically, the exclusive focus on transformer-based autoregressive models ignores significant portions of the neural architecture landscape, including encoder-decoder models (T5), bidirectional models (BERT), state-space models (Mamba), and recurrent architectures that may show fundamentally different processing patterns.

### Biological Disconnect Critical for NeurIPS

The extended abstract operates in complete isolation from cognitive neuroscience literature, representing a fundamental limitation for NeurIPS where computational-biological connections are central. The work resembles ``a child reading exam instructions and responding'' but provides no investigation of whether this replicates human brain processes or represents computational artifacts specific to transformers. Existing neuroscience literature on instruction processing phases (P300/P600 ERP components, frontal-parietal network timing) remains completely unaddressed, missing opportunities to validate or contextualize findings within broader cognitive science.

### Methodological Oversimplification

The activation patching approach, while mathematically sound, makes critical simplifying assumptions analogous to ``physics without air resistance.'' The formula
\begin{equation}
\tilde{h}_i^{(\ell)} = \begin{cases}
\mathbf{h}_i^{(\ell)}(S), & i \in \mathcal{M} \\
\mathbf{h}_i^{(\ell)}(B), & i \notin \mathcal{M}
\end{cases}
\end{equation}
ignores token interaction effects, activation scale normalization, temporal dynamics, and global context effects that likely influence real neural processing.

### Limited Task Complexity and Scope

The evaluation focuses exclusively on simple, direct-lookup tasks without complex reasoning, mathematical problem-solving, or creative generation. While multi-hop tasks are included, they remain within the simple compositional domain. The generalizability to cognitively demanding tasks that better reflect real-world language model usage remains undemonstrated.

### Structural and Clarity Issues

The document requires multiple readings to fully understand the experimental structure, particularly due structure. The methodology would benefit from visual flowcharts, graphical diagrams and clearer algorithmic presentation, even showing off the employed pipeline.

# Quality: 3 (Good)

The work is technically sound with appropriate methodology and well-supported claims within its scope. The experimental design effectively isolates the phenomenon of interest, and results are consistent across tested conditions. However, the severely limited model coverage and methodological simplifications prevent a higher rating. The authors are reasonably honest about scope limitations but could be more explicit about generalizability constraints.

# Clarity: 3 (Good)

The document is generally well-written and organized, though requires re-reading for full comprehension. The core concepts are ultimately accessible, but the experimental structure could be presented more clearly. The mathematical formulation is appropriate, and sufficient detail is provided for reproduction by expert readers.

# Significance: 3 (Good)

For the transformer interpretability community, this represents potentially excellent work (4) providing the first systematic framework for localizing cognitive transitions. However, for NeurIPS specifically, the complete disconnection from neuroscience and overstated claims about ``LLM'' generality significantly limit broader impact. The multi-hop findings strengthen significance by demonstrating onset timing independence from task complexity, suggesting fundamental rather than superficial phenomena. Personally, I consider the work gives a step on the right direction, yet due length, it does not show or expose questions or variables relevant to the topic.

# Originality: 4 (Excellent)

The work provides genuinely novel insights by first systematically localizing the comprehension$\rightarrow$execution transition in neural networks. The ``onset'' concept with clear operational definition represents significant conceptual innovation. The application of activation patching to this fundamental question about cognitive transitions is original and well-motivated. The discovery that multi-hop complexity doesn't shift onset timing provides new theoretical insights about instruction processing in neural models.

# Questions

## 1. Cross-Architectural Validation and Biological Plausibility

The exclusive focus on autoregressive transformers raises critical questions about generalizability and biological relevance. How would the onset phenomenon manifest in more biologically plausible architectures like recurrent networks or spiking neural networks? Given that biological attention mechanisms differ fundamentally from transformer self-attention, and cortical processing involves parallel streams rather than sequential layers, are transformers appropriate models for investigating phenomena intended to inform understanding of cognitive processes?

_Evaluation Impact_: Addressing cross-architectural validation or providing theoretical justification for transformer choice could increase significance rating to 4.

## 2. Mechanistic Understanding of Onset Transition

While they effectively identify \textit{where} the onset occurs, it provides no analysis of \textit{what} computational operations change at this transition point. What specific mechanisms drive the shift from flexible processing to fixed execution? How do attention patterns, token representations, or computational pathways reorganize around the onset layer?

_Evaluation Impact_: Even preliminary mechanistic analysis could increase quality rating and provide more actionable insights for the field.

## 3. Biological Correspondence and Neuroscience Integration

Given NeurIPS's mission to bridge computational and biological neural systems, how do these findings relate to known neuroscience of instruction processing? The temporal separation found in ERP studies (P300/P600 components) and fMRI evidence of distinct frontal-parietal activation phases during instruction comprehension vs. execution suggest biological analogs may exist. Could the authors connect their computational findings to this literature?

_Evaluation Impact_: Adding even speculative biological discussion could significantly boost significance for NeurIPS audience and more researchers on the field. In fact, this should be a high priority focus for the paper in the NeurIPS context.

## 4. Real-World Task Generalization

The evaluation focuses on simple, direct-lookup tasks. How does onset timing change for complex reasoning tasks, mathematical problem-solving, or creative generation that better reflect practical language model applications? Do the findings hold for tasks requiring extended reasoning chains, ambiguous instructions, or conflicting requirements?

_Evaluation Impact_: Demonstrating generalization to complex tasks would strengthen both quality and significance ratings.

##  5. Methodological Robustness and Token-Level Analysis

The current approach treats token masks as atomic units but provides no analysis of individual token importance or interaction effects. How does onset timing vary when patching individual tokens versus token groups? Are certain instruction tokens more causally important for onset timing than others?

_Evaluation Impact_: More granular analysis could address methodological limitations and increase quality rating.

# Limitations

### No. The authors do not adequately address limitations:

- **Architectural Specificity:** No discussion of how findings may be specific to autoregressive transformers rather than neural computation generally
- **Biological Relevance:** No acknowledgment that transformer architectures may be poorly suited for investigating biological cognitive processes
- **Task Scope Limitations:** Insufficient discussion of generalization constraints to simple task types
- **Methodological Simplifications:** No acknowledgment of interaction effects, scaling issues, or temporal dynamics ignored by the patching approach

# Recommendations:
Avoid using exaggerated highlights or claims given the narrow scope worked. Add a section addressing architectural constraints, biological relevance or comparison, task scope limitations, and methodological assumptions. While the work represents solid technical contribution to transformer interpretability, does not focus on the core of NeurIPS: bridging computational and biological neural
systems.

**Score:**

3

**Topic Fit:**

2

---

### Official Review · Reviewer_kxuq · 2025-09-16
**Official Comment**

**Confidence:** 4

**Review:**

The paper investigates instruction onset in transformer-based LLMs, particularly Llama models. The authors ask: at what layer does the model transition from passively reading a prompt to actively executing the instruction? To study this, they design three synthetic datasets (Key–Value, Quote Attribution, and Letter Selection) and their multi-hop extensions. Using activation patching on minimally altered prompts, they compute a “flip rate” that reflects causal dependence on instruction tokens. Results consistently reveal an inflection point, termed instruction onset, where interventions on instruction tokens cease to alter the outcome. Multi-hop variants show that additional procedural complexity does not push the onset deeper. Late-context analysis further suggests weaker, compressed consolidation at the final token.
## Strengths
1. The paper reframes instruction following as a layer-localized onset event, offering a measurable coordinate within the model rather than treating instruction following as an all-or-nothing property.
2. Activation patching on contrastive, token-aligned prompts provides a clean causal probe, isolating instruction-specific effects.
3. The design of three synthetic datasets, with carefully constructed counterfactuals, ensures interpretability and avoids confounds from world knowledge.
4. Instruction onset is observed across different Llama model sizes, highlighting the robustness of the phenomenon.
5. Demonstrates that task complexity does not systematically shift onset, suggesting that instruction interpretation stabilizes early and execution dominates thereafter.
## Weakness:
1.  Results are limited to the Llama family. Extending to other model families, such as Mistral or Qwen, would strengthen claims of generality.
 2. The datasets are narrow, single-token-answer tasks, raising concerns about ecological validity. Real-world instruction following often requires multi-token reasoning, memory, or factual recall.
 3. The strong conclusions may be overstated given the reduced setting. Broader tasks could reveal different or more nuanced onset dynamics.

## Comments:
 1. In Figure 1, the progression of transformer layers on the x-axis could be labeled more explicitly to better illustrate onset progression.
 2. The FlipRate equation omits a definition for N (number of data points), which could confuse readers.

**Score:**

3

**Topic Fit:**

3

---

### Official Review · Reviewer_rExH · 2025-09-16
**This work analyzed the starting point of instruction following (onset) inside an LLM network using the three purpose-built datasets introduced in the paper.**

**Confidence:** 4

**Review:**

The method proposed in the paper is innovative, and the results give interesting insights into the location (transformer layer) of instruction onset across different Llama models investigated in this study.

Some follow up comments:

1.	It would be helpful to add a little more context as to why the content-control patching result in zero flip-rate across all layers.

2.	As this study only investigated Llama models, it would be interesting to see if similar conclusions can be drawn across other LLM families.

**Score:**

4

**Topic Fit:**

2